# Evaluation of Thermal Effects on the Bioactivity of Curcumin Microencapsulated with Porous Starch-Based Wall Material Using Spray Drying

**Chenwei Huang [1], Shengwen Wang [2] and Huaiwen Yang [2],***

[1]   Center of Academia-Industry Collaboration and Continuing Education Innovation Incubation, National Chiayi University, Chiayi City 60004, Taiwan; discovery@mail.ncyu.edu.tw

[2]   Department of Food Science, National Chiayi University, Chiayi City 60004, Taiwan; KevinWang790116@gmail.com

*   Correspondence: calyang@g.ncyu.edu.tw; Tel.: +886-5-271-7620

**Abstract:** Curcumin was microencapsulated by porous starch using a spray dryer with a particle size between 1.5 and 2.0 μm and subjected to water bath (40–100 °C) and oven heating (150–200 °C) in comparison to non-encapsulated samples. The minimum possible encapsulation rate ranged from 26.75 to 52.23%. A reasonable thermal stability was observed after water bath heating with regard to 2,2-diphenyl-1-picrylhydrazyl free radical (DPPH) scavenging activity. On the other hand, the increase in oven heating temperature caused significant alterations compared with the control samples ($p < 0.05$). The encapsulated particles subjected to oven heating at 170 °C demonstrated serious collapse. The DPPH scavenging activity of non-encapsulated curcumin was significantly reduced ($p < 0.05$) from 48.94% ± 3.72% (control, 0 °C) to 40.42% ± 2.23% (oven heating, 160 °C); however, remained stable for the encapsulated samples (51.18% ± 4.86%–50.02% ± 1.79%) without significant difference ($p < 0.05$). The ABTS scavenging activity was promoted as a function of the oven heating temperature. Both DPPH and ABTS free radical scavenging activities remained stable after water bath. Nevertheless, the color of microencapsulated curcumin was better preserved in comparison to the controls.

**Keywords:** encapsulated curcumin; spray drying; free radical scavenging activity; color attribute; oven heating

## 1. Introduction

Turmeric (*Curcuma longa*) is a perennial herb, and the yellow powder obtained upon grinding its rhizomes has long since been consumed by humans and its usage is prevalent in several countries [1]. Turmeric powder is one of the main spices used in curry and acts as a coloring agent. Curcumin, also known as diferuloylmethane, is the main active and coloring component in turmeric powder. Several studies have shown that curcumin can prevent chronic diseases by trapping free radicals as well as by exerting its antioxidant effects. However, curcumin is easily destroyed by heating [1,2].

Microencapsulation was first used in the manufacturing of carbon paper, before subsequently being used in textiles, medicine, cosmetics, and food processing [3–5]. Microencapsulation refers to the technique of encapsulating core material with a high-molecular-weight wall material to form microcapsules. The main aim of this technique is to use the wall material for separating the core material from the outside environment to envelop, protect, store, or release drugs [6,7]. Microencapsulation has gained attention because wall materials act as a barrier against light, heat, acid, base, and humidity. In addition, wall materials can suitably separate reactants to prevent immediate oxidation or hydrolysis and conditionally release the core material at suitable timepoints for facilitating its reaction with

reactants [7,8]. Microencapsulation is gradually being used in the food industry, where it can be employed to encapsulate food additives (acidulants, flavoring agents, seasoning agents, or sweeteners), probiotics, and coloring agents. The use of microencapsulation to process coloring agents can improve the solubility and stability of these agents [9–13].

Spray drying, also known as bi-fluid drying, is a method that uses high pressure gas to atomize liquids into a mist before convection is used to immediately evaporate the water in the mist droplets. The convection effects of hot air are used to vaporize the moisture in the liquid, thereby facilitating the drying of the solids in the solution, and this method of instant vaporization of water from liquids using hot air is termed as spray drying. Spray drying is often employed to prepare microcapsules with multi-core structure using the pores formed owing to the penetration of wall materials by core materials, which further increase in size after spray drying. The core materials then occupy these pores and are dispersed between the wall material and core sections [7,14–16]. This multi-core structure can be confirmed via microscopic observation.

In a recent study, microencapsulated curcumin was evidenced to improve stability while different dry methods (spray and lyophilization) were facilitated [17]; the results indicated that spray-dried curcumin capsules possessed a higher retention rate under light exposure during long term storage comparing to lyophilized ones. Such results make spray-dried curcumin capsules appealing for further study regarding the wall material formulation including the employments of *Aloe vera* mulcilage, casein, kappa-carrageenan, and casein and the antioxidant properties the capsules were also evaluated [18–20]. With the aid of rhamnolipids as a biosurfactant (emulsifier), curcumin was also successfully facilitated into a liposomal delivery system in the light of nutra/pharmaceutical preparation even with only an ethanol-injection method [21]. However, to the best of our knowledge, currently, there is no study regarding the use of spray drying to encapsulate curcumin and assessment of its antioxidant activity as well as the tolerability of its coloring component upon thermal processes simulating culinary arts. Therefore, the present study aimed to use β-cyclodextrin and porous starch as wall materials for the microencapsulation of curcumin with proper emulsification and thickening agents. An orthogonal experimental design was used to optimize the microencapsulated curcumin (MEC) prepared by spray drying with regards to the encapsulated rate and particle size. The changes in its stability and bioactivity under common food thermal conditions were assessed to improve its applicability in food processing.

## 2. Materials and Methods

### 2.1. Materials

Sample powder (of Indian origin) containing 95% curcumin, which was purchased from Bo-Cheng flavoring ingredient Inc., Tainan, Taiwan, was employed as the MEC core material. β-cyclodextrin, potato starch, sucrose stearic acid ester S-1170, and sodium carboxymethyl cellulose (CMC) were food grade and obtained from local sources. The antioxidant standards—di(phenyl)-(2,4,6-trinitrophenyl) iminoazanium (DPPH), 2,2'-azino-bis(3-ethyl benz-thiazo line-6-sulphonic acid) (ABTS), and 6-Hydroxy-2,5,7,8-tetramethylchroman-2-carboxylic acid (Trolox) were purchased from Sigma-Aldrich GmbH (Sternheim, Germany). The high-performance liquid chromatography (HPLC)-grade methanol was used for curcumin analysis; all other chemicals used were analytical grade obtained from either Sigma-Aldrich or Merck (Darmatadt, Germany).

### 2.2. Experimental Framework

In the present study, a three-level orthogonal experimental design was employed that was based on the four controlling factors for emulsification—wall material percentage, excipient weight, homogenization temperature, and homogenization time—to identify the optimal microencapsulation rates for subsequent heat resistance analyses. The similar orthogonal array have been employed microencapsulation studies on curcumin and allicin utilizing spray drying technique [22,23]. Such an

experimental framework is denoted as an L9($3^4$) center composite array, 9 runs involved in 3 levels for 4 factors with systematic vector of [−1, 0, 1]. The vector indices were marked in Table 1 with parentheses. Changes in the curcumin content and antioxidant activity (DPPH and ABTS free radical scavenging activities), as well as the changes in color before and after simulated thermal processing, were examined.

**Table 1.** L9($3^4$) § Orthogonal array of the present study with minimum encapsulation efficiency and viscosity of curcumin containing emulsion.

| Exp.** | Factors * | | | |
|---|---|---|---|---|
| | Wall Material (%) | Excipient (g) | HMG Temperature (°C) | HMG Time (h) |
| 1/(A) | 1 (−1) | 0.1 (−1) | 40 (−1) | 1 (−1) |
| 2/(B) | 1 (−1) | 0.15 (0) | 50 (0) | 1.5 (0) |
| 3/(C) | 1 (−1) | 0.2 (1) | 60 (1) | 2 (1) |
| 4/(D) | 3 (0) | 0.1 (−1) | 50 (0) | 2 (1) |
| 5/(E) | 3 (0) | 0.15 (0) | 60 (1) | 1 (−1) |
| 6/(F) | 3 (0) | 0.2 (1) | 40 (−1) | 1.5 (0) |
| 7/(G) | 5 (1) | 0.1 (−1) | 60 (1) | 1.5 (0) |
| 8/(H) | 5 (1) | 0.15 (0) | 40 (−1) | 2 (1) |
| 9/(I) | 5 (1) | 0.2 (1) | 50 (0) | 1 (−1) |

* HMG: Homogenization. ** Exp.: Experimental sample. § 9 runs involved in 3 levels for 4 factors. (A) through (I) refer to the representation of Figure 1.

### 2.3. Production of MEC

#### 2.3.1. Operation of Spray Dryer

The wall material and MEC were prepared using the spray dryer (Pulvis Mini Spray Model GA32, Yamato Scientific Ltd., Tokyo, Japan), according to a method described previously [23]. The conditions used were as follows: feed velocity = 10 mL/min, air speed = 40 m³/h, and inlet temperature = 120 °C. The volume of each spray drying sample was 900 mL, and spray drying duration was 1.5 h. After samples were used to prepare MEC powder, they were stored at −20 °C until further use.

#### 2.3.2. Preparation of the Wall–Core Emulsion

Regarding the preparation of porous starch, according to the method described by Chang, Yu, and Ma [24], 5 g of potato starch were added to 100 mL of distilled water. The mixture was heated to 95 °C and maintained for 30 min until complete gelatinization was achieved. The potato starch suspension was stored at 5 °C for 48 h until it solidified into a gel. The gel was cut into 1-cm³ blocks and stored in a −10 °C freezer for 48 h. The frozen gel blocks were immersed in 95% edible alcohol at room temperature for 3 h before placing them in a convection-heating oven for 6 h at 50 °C and followed by 2 h of drying at 100 °C until alcohol was completely removed to obtain porous starch.

For the preparation of the wall–core emulsion, the prepared porous starch and β-cyclodextrin were mixed in a 7:1 ratio and dissolved in distilled water at ≥70 °C to obtain an aqueous solution; to obtain a thickened emulsion, 20 mg of curcumin (dissolved in 95% alcohol), 1–5% *w/v* wall material powder, and 1–2 g excipient were dissolved in deionized water before heating the mixture in a water bath (40–60 °C), along with high-speed homogenization at 4000 rpm (1–2 h) using a Polytron RT-3000 homogenizer (Kinematica Inc., Luzern, Switzerland). The excipient comprised equal parts of sucrose stearic acid ester (S-1170) and CMC, with S-1170 as emulsification agent and CMC as thickener. The viscosity of the emulsion, which would determine the subsequent encapsulation rate, was measured using a viscometer (DV-II+ Brookfield Engineering Laboratories, Inc., Middleboro, MA, US) and a RV-4 spindle at a rotational speed of 100 rpm. The controlling factors, including the wall material percentage, excipient weight, water bath temperature, and homogenization time, were used

to prepare nine samples, as listed in Table 1, according to the three-level orthogonal experimental design before spray drying to form MEC. Every batch of 900 mL of the thickened emulsion matrix was subjected to spray drying to form MEC.

Four calibration curves were made for curcumin measurements:(I) 20 mg 95% curcumin alone, (II) 20 mg 95% curcumin + 0.85 g wall material, (III) 20 mg 95% curcumin + 0.2.45 g wall material, and (IV) 20 mg 95% curcumin + 4.05 g wall material. Each group was reconstituted in 10 mL of methanol for 30 min before extraction was performed using an ultrasonic water bath ($47 \pm 3$ kHz with power level of 120 W). Thereafter, the sample was centrifuged for 15 min at 3000 rpm using a refrigerated centrifuge (Avanti J-15, Beckman Coulter Inc., Indianapolis, IN, US). The supernatant was collected and a serial dilution was prepared (10, 5, 2.5, 1.25, and 0.625 ppm). Subsequently, an absorbance wavelength ($\lambda$) of 420 nm was set to measure the absorbance. The standard concentration curve for curcumin was obtained after performing a regression with a $R^2$ greater than 0.9993. The regression curves were used for corresponding samples with different incorporated wall material.

## 2.4. Characteristics of MEC

### 2.4.1. Particle Size Evaluation of MEC

A transmission electron microscope with photography function (BX-51, Olympus, Tokyo, Japan) was used to observe the particle size of MECs (at 100× magnification). The scale was used to estimate the mean particle size of one MEC.

### 2.4.2. Quantitation of Curcumin Content in Every Group of Spray-Dried MEC Powder

Spray-dried MEC powder (20 mg) from nine experimental samples was weighed on a 5 ppm basis and reconstituted in 10-mL methanol, followed by extraction in the ultrasonic water bath and centrifugation using the refrigerated centrifuge as describe before. The corresponding standard curve was used to measure curcumin content.

### 2.4.3. Measurement of Curcumin Encapsulation Rate of Spray-Dried MEC for Each Group

The concentration of spray-dried MEC powder of every group was converted from ppm to g before the minimum possible encapsulation rate was calculated using the following formula:

$$minimum\ ER\ (\%) = \frac{M_m}{M_t} \times \frac{M_f}{M_i} 100 \tag{1}$$

where $M_t$ refers to total added curcumin mass and $M_m$ refers to the mass in the microcapsules; $M_f$ refers to the final mass of the microcapsule solids after spray-drying and $M_i$ refers to the initial mass of the raw microcapsule solids.

Thereafter, the orthogonal table was used for factor analysis to identify the most suitable encapsulation rate factor and samples from three groups with better encapsulation rates were selected for the heat resistance analysis of spray-dried MEC.

## 2.5. Heat Resistance Analysis of Spray-Dried MEC

### 2.5.1. Heating Simulation

For hydrothermal cooking, samples were precisely weighed and reconstituted in methanol before subjecting them to water bath processing at 50–100 °C (an increment of 10 °C with 10 min of heating), which simulated the normal hydrate cooking temperature and duration of foods. Oven heating at 150–200 °C (increment/time = 10 °C/30 min) was the simulated temperature and duration for baked products. During oven heating, each sample was placed in the same position in the oven for avoiding temperature differences. Unheated (0 °C) samples were used as the control for the water bath and oven heating samples. After heating, ultrasonic water bath was used for 30 min of extraction, followed

by centrifugation at 3000 rpm for 15 min in a refrigerated centrifuge. The supernatant was collected for heat resistance analysis.

### 2.5.2. Heat Resistance Analysis

Items of this analysis for the water bath samples included changes in curcumin content and in DPPH and ABTS free radical scavenging activities. As for the evaluations of DPPH and ABTS free radical scavenging activities, the corresponding standard curves established in Section 2.4.2 were used to decide the sample weight needed for methanol reconstitution to a 50 ppm concentration level. In addition, color analysis was included for the oven heating samples. Regarding the curcumin content, the standard curve and measurement method from Sections 2.4.2 and 2.4.3 were used to evaluate residual curcumin content following water bath heating and oven heating.

By modifying the method of Jafari et al. [25], the sample supernatants were serially diluted to the concentrations of 50, 25, and 12.5 ppm. 0.05 mL of samples were added to a 96-well plate before adding 0.1 mM DPPH (0.24 mL) and these were thoroughly mixed. The plates were incubated in the dark for 50 min before measuring absorbance at λ = 517 nm using a microplate reader (Spectra Max 190, Molecular Devices Inc., San Jose, CA, US). The following formula was used to calculate the DPPH free radical scavenging activity of the samples. Trolox was used as the standard, and methanol was used for the control.

$$DPPH\ \%\ Inh = \left(\frac{A_0 - A_1}{A_0}\right) \times 100 \tag{2}$$

where: $A_0$ is the absorbance of the standard and $A_1$ is the absorbance of the sample.

The method of Jafari et al. [25] was modified and 7 mM ABTS$^+$ solution was added to 2.45 mM potassium persulfate solution (1:1 mixing and incubating in a dark room for 12–16 h). Following this, 95% ethanol was used for dilution, and absorbance at λ = 734 nm was adjusted to 0.7 ± 0.02. The sample supernatant was also serially diluted to 50, 25, and 12.5 ppm, and 8 μL of sample was added to a 96-well plate before adding 0.24 mL of ABTS diluent. After incubation in the dark room for 6 min, the absorbance at λ = 734 nm was measured using a microplate reader, and the following formula was used to calculate the ABTS free radical scavenging activity of the sample. Trolox was used as the standard, and methanol was used for the blank control.

$$ABTS\ scavenging\ activity\ (\%) = \left(\frac{AB - AA}{AB}\right) \times 100 \tag{3}$$

where: $AB$ is the absorbance of ABTS radical + methanol; $AA$ is the absorbance of ABTS radical +s ample extract/standard.

For color analysis, a colorimeter (ZE-2000, Nippon Denshoku Ltd., Tokyo, Japan) was used to measure changes in the L, a, and b values of spray-dried curcumin MEC powder before and after oven heating according to the method of Barreiro, Milano, and Sandoval [26]. First, a standard whiteboard (Y = 93.73, X = 95.61, and Z = 113.46) was used for correction.

### 2.6. Statistical Analysis

The Statistical Package for the Social Science (SPSS version 19.0) was used for statistical analysis. One-way analysis of variance was used for significance, and Duncan's multiple range test was used for comparison of significant difference. The Pearson correlation analysis between color parameters and antioxidant activity was also conducted. A difference of $p < 0.05$ was considered significant.

## 3. Results and Discussion

### 3.1. Particle Size Analysis of MEC

Figure 1 shows the observed particle size of experimental samples 1–9 under the transmission electron microscope at a magnification of 100×. The particle size was found to be 1.5–2.0 μm.

Some particles underwent aggregation, and the aggregated particles in Figure 1B,C are larger, whereas others in Figure 1A through Figure 1I are smaller. The aggregate particle size in Figure 1 can be classified as follows: 1B = 1C > 1E = 1D = 1A > 1F = 1H = 1G = 1I. As curcumin itself is hydrophobic and viscous, this finding may be associated with the subsequent encapsulation rate. Typically, the particle size of spray-dried powder is <10 μm [24]. According to the manufacturer instructions for the spray dryer used in the present study, particle size from the atomizing nozzle should be 2.0–3.5 μm. The partial aggregation phenomenon shown in Figure 1 may be attributable to the storage of the spray-dried powder in a −4 °C freezer and its thawing at room temperature that resulted in water vapor causing the powder to form aggregates. Therefore, this phenomenon was observed under transmission electron microscope.

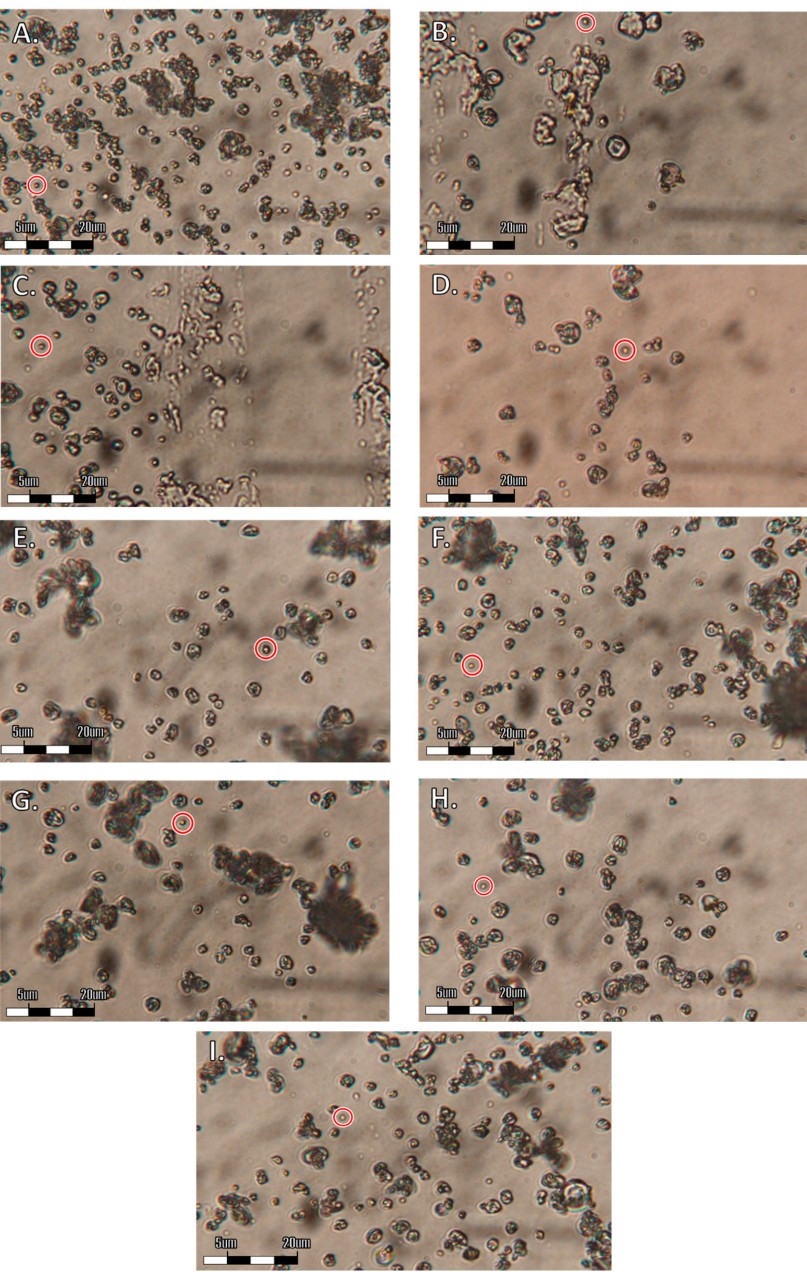

**Figure 1.** The graphic images of microencapsulated curcumin (MECs) (5 % of wall material contained in S7–S9, i.e., (**G**–**I**) on a 5 ppm curcumin content bases) observed under transmission electron microscope (100× magnification) corresponding to the sample number listed in Table 1 (Experimental samples 1 → 9 referred to (**A** → **I**)).

In addition, the particle size was smaller than that achieved using the atomizing nozzle (2.0–3.5 µm) owing to the effects of emulsion viscosity, solid content, and other physical characteristics of the emulsion. Furthermore, the settings of the spray dryer have an influence on the particle size: a higher inlet temperature results in faster drying and smaller particle size. When loading speed is fast and drying remains incomplete, particles would aggregate and the powder particles from the spray dryer would be larger. In addition, the type of atomizing nozzle affects the powder particle size. Typically, an atomizing nozzle that mixes high pressure gas and sample provides better results than high-speed rotating blades; moreover, the spray-dried powder particle size is smaller [24].

### 3.2. MEC Encapsulation Rate Analysis

Table 2 lists the minimum possible encapsulation rates. Despite different ratio between wall and core materials, robust calibration curves were obtained as described in Section 2.4.2. The experiment results showed that the minimum possible encapsulation rate ranged from 26.75 to 52.23%, which is consistent with the results of spray drying studies on curcumin pigment and allicin, a diallyl thiosulfinate mainly derive from garlic [22,23]. Additionally, these results are consistent with the results if encapsulation rates reported previously [27].

**Table 2.** The encapsulation efficiency and viscosity of curcumin containing emulsion.

| Exp. | Measurement | |
|------|-------------|---|
| | Minimum ER * (%) | Viscosity ** (cPoise) |
| 1 (A) | 29.8 ± 1.44 [ab] | 32 ± 2.1 [a] |
| 2 (B) | 27.4 ± 2.61 [a] | 34 ± 1.7 [ab] |
| 3 (C) | 26.8 ± 1.14 [a] | 44 ± 0.8 [d] |
| 4 (D) | 44.1 ± 1.53 [d] | 36 ± 1.21 [b] |
| 5 (E) | 32.3 ± 0.91 [bc] | 40 ± 1.50 [c] |
| 6 (F) | 34.7 ± 1.08 [c] | 48 ± 2.06 [e] |
| 7 (G) | 45.9 ± 1.25 [d] | 48 ± 1.46 [e] |
| 8 (H) | 44.9 ± 3.45 [d] | 48 ± 2.14 [e] |
| 9 (I) | 52.2 ± 2.19 [e] | 56 ± 1.49 [f] |

* ER: Minimum encapsulation rate. ** For viscosity measurement, the temperature was between 23.5 °C and 29.5 °C and the rotational speed was 100 rpm. All values are means of triplicates. Means followed by lowercase letters in the same column are significantly different due to thermal treatments ($p < 0.05$).

In addition, the ratio of core material and wall material would affect the encapsulation rate. For example, when the ratio of core material to wall material is 1:4, the encapsulation rate should be approximately 20% while aqueous solubility and stability of curcumin was studied [28]. In the present study, the ratios of core material to wall material was 1:25, 1:42.5, and 1:122.5 and the much higher encapsulation rate can be attributed to the higher ratio of wall material.

From the orthogonal array, which evaluated the effects of different factors, the ratio of wall material was found to be the greatest factor that affects the encapsulation rate. The results showed that encapsulation rate typically increased when the ratio of wall material increased. Reportedly, the physicochemical characteristics of wall material are one of the main factors that determine the encapsulation rate [22]. β-cyclodextrin and porous starch were used as microencapsulation wall materials in the present study. In the literature, it has been reported that β-cyclodextrin can protect volatile substances from damage during heating in the spray drying process [29]. In addition, β-cyclodextrin has a suitable molecule size and exhibits good biocompatibility, and it has a hydrophobic interior and hydrophilic exterior. These structural characteristics can tightly absorb hydrophobic curcumin [30]. Due to its honeycomb structure and high emulsification ability, porous starch can help β-cyclodextrin form a tight mesh structure to increase the absorption and adhesion, ensuring that curcumin is sufficiently encapsulated in the wall material, thereby resulting in a higher encapsulation rate [31].

Spray drying involves the evaporative cooling effect. This phenomenon is based on the principle that when air and liquid solution come in contact and mix, water droplets will absorb heat from air and form water vapor. Therefore, the heat content of air will decrease, inherently decreasing the temperature. However, as water vapor increases, the relative humidity of air also increases [32]. Therefore, the choice of inlet temperature and hot air flow speed would have an effect on spray drying to some extent. We selected a spray drying temperature of 120 °C because previous research has shown that volatile substances do not easily volatilize at low temperatures, resulting in incomplete spray drying and subsequent adherence of the particles to the spray drying chamber, thereby decreasing the yield. However, excessively high temperature will prohibit the bioactivity of the spray-dried sample [22]. Related studies have shown that in downstream spray drying at 120 °C, the core temperature of MEC is <100 °C [26], which is the phenomenon caused by the evaporative cooling effect as well. We selected a spray drying rate of 40 m$^3$/h because an excessively low flow rate will result in an inability to completely separate MEC and dry air flow in a cyclone separator [26]. In addition, the increased relative humidity due to the evaporative cooling effect becomes unfavorable for the drying air flow to convey atomized suspension to the cyclone separator, causing more samples to adhere between the drying chamber and cyclone separator and thereby decreasing the yield. An excessively high flow rate will cause MECs to be carried by the drying airflow away from the spray dryer. In the literature, although it has been mentioned that the effects of the flow rate are not significant, we believe that it would result in some loss [26].

The practical inconvenience of spray drying is that extremely viscous substances tend to block the atomizing nozzle, resulting in discontinuous production. Furthermore, emulsion viscosity is reportedly a factor that affects microencapsulation rate [26]. Therefore, we performed viscosity analysis of the emulsion before spray drying in the present study under the ambient temperature range (23.5–29.5 °C), and the obtained viscosity range was 34–56 cp. This viscosity range does not cause any obstruction at the atomizing nozzle during spray drying and can achieve good encapsulation rates and hence can be used as a reference for future studies. Table 2 shows the viscosity of various samples. Samples 6–9 having higher encapsulation rate showed higher viscosity as well, proving that viscosity can affect the encapsulation rate.

Experimental samples 7–9 were observed to have considerably superior minimum encapsulation rates, and the orthogonal array results showed that the factor that greatly affects the encapsulation rate is the ratio of wall material. Therefore, experimental samples 7–9 that were subjected to MEC treatment were named as S7-S9, respectively, and as heat resistant study samples to facilitate subsequent results and discussion; whereas, a non-MEC (control) was also prepared accordingly without the incorporation of the wall material and excipient. For the best possible minimum ER, the 0.2 g of excipient, 50 °C of HMG temperature, and one hour of HMG time would be considered as an optimal options.

*3.3. Effects of Heating Type and Temperature*

3.3.1. Changes in Curcumin Content

Table 3 also shows the changes in curcumin content in non-MEC control and S7–S9 at 0 °C and 50–100 °C of water bath as well as 150–200 °C of oven heating. When the control was hydrothermally heated to 100 °C, its curcumin content decreased before slightly increasing to 6.24 ± 0.00 ppm, which was consistent with the findings in the literature that high temperature would occasionally cause the increase curcumin solubility even though not drastically [32]. The presumed content that directly converts from the original potency is approximately 5 ppm. We noticed that the resulted measured value of non-MEC control is higher than those of MEC ones because the non-MEC control did not include the wall material and excipient. Therefore, the methanol reconstitution of the non-MEC control could be more efficient comparing to those of MEC ones.

**Table 3.** The changes of curcumin content in MECs (5% of wall material contained in S7–S9 on a 5 ppm curcumin content bases) after water bath and oven heating using different processing temperatures.

| Heating Method | Temperature (°C) | Experimental Sample | | | |
|---|---|---|---|---|---|
| | | Control | S7 | S8 | S9 |
| Water Bath | 0 | 6.91 ± 0.00 [Aa] | 4.40 ± 0.00 [Ec] | 4.49 ± 0.01 [Fb] | 4.33 ± 0.00 [Bd] |
| | 50 | 5.89 ± 0.01 [Fa] | 4.49 ± 0.00 [Dc] | 4.63 ± 0.01 [Db] | 4.34 ± 0.00 [Bd] |
| | 60 | 5.96 ± 0.00 [Ca] | 4.62 ± 0.00 [Ab] | 4.55 ± 0.00 [Ec] | 4.25 ± 0.01 [Dd] |
| | 70 | 5.91 ± 0.01 [Da] | 4.55 ± 0.00 [Bc] | 4.73 ± 0.00 [Bb] | 4.25 ± 0.01 [Dd] |
| | 80 | 5.85 ± 0.00 [Ga] | 4.37 ± 0.00 [Fc] | 4.67 ± 0.01 [Cb] | 4.30 ± 0.00 [Cd] |
| | 90 | 5.89 ± 0.00 [EFa] | 4.37 ± 0.00 [Fc] | 4.43 ± 0.01 [Gb] | 4.31 ± 0.01 [Cd] |
| | 100 | 6.24 ± 0.00 [Ba] | 4.55 ± 0.00 [Bc] | 4.81 ± 0.00 [Ab] | 4.24 ± 0.01 [Dd] |
| Oven Heating | 0 | 6.91 ± 0.00 [Aa] | 4.40 ± 0.00 [Ac] | 4.49 ± 0.01 [Ab] | 4.33 ± 0.00 [Ad] |
| | 150 | 2.61 ± 0.01 [Ba] | 2.46 ± 0.00 [Bb] | 2.45 ± 0.00 [Bb] | 2.24 ± 0.00 [Bc] |
| | 160 | 2.14 ± 0.00 [Cb] | 2.08 ± 0.00 [Cc] | 2.26 ± 0.01 [Ca] | 1.85 ± 0.01 [Cd] |
| | 170 | n.a. | 1.60 ± 0.00 [Da] | 1.30 ± 0.00 [Db] | 1.26 ± 0.01 [Dc] |
| | 180 | n.a. | 1.29 ± 0.00 [Ea] | 0.99 ± 0.00 [Eb] | 0.95 ± 0.00 [Ec] |
| | 190 | n.a. | 1.05 ± 0.00 [Fa] | 0.85 ± 0.00 [Fb] | 0.76 ± 0.02 [Fc] |
| | 200 | n.a. | 0.86 ± 0.01 [Ga] | 0.55 ± 0.01 [Gc] | 0.67 ± 0.00 [Gb] |

All values are means of triplicates. Means bear capital letters within the same column are significantly different due to temperature ($p < 0.05$). Means bear lowercase letters in the same row are significantly different due to samples ($p < 0.05$). n.a.: data not available.

### 3.3.2. Changes in DPPH Free Radical Scavenging Activity Due to Water Bath

Figure 2 shows the changes in DPPH free radical scavenging activity of the control and S7–S9 at 50 ppm after heating for 10 min under different water bath temperatures. Based on the results shown in Figure 2, there were no significant differences in DPPH free radical scavenging activity in the samples from 0 to 100 °C ($p \geq 0.05$); although DPPH free radical scavenging activity began to decrease at 50 °C (from 47.25% ± 2.30% to 38.18% ± 2.43%) in S9, it increased again from 38.18% ± 2.43% to 49.09% ± 1.19% after temperature increased to >100 °C. Because the results of control and the literature [25] showed that there was no significant difference in DPPH when the control was heated for 10 min at 50–100 °C ($p \geq 0.05$) and curcumin absorbance was only decreased by 6.2% after heating at 100 °C for 10 min compared with the 0 °C experiment, the possibility that S9 exhibits lower DPPH free radical scavenging activity at 50–80 °C due to curcumin degradation can be excluded. Therefore, it can be inferred that S9 has a better encapsulation rate and that heating causes the mesh structure formed between β-cyclodextrin and porous starch to form tighter links with curcumin. This causes incomplete dissolution when methanol was used for reconstitution as well as a decrease in DPPH free radical scavenging activity. High temperature (100 °C) disrupts this mesh structure, releasing curcumin, which increases the DPPH free radical scavenging activity.

Overall, the DPPH free-radical scavenging activity of curcumin did not undergo significant changes when heated to 100 °C. This finding answers the question from the previous encapsulation rate result regarding heat destruction due to spray drying. In addition, no significant changes were noted between the control and S7–S9. Furthermore, in the literature, it has been shown that curcumin exhibits better thermal stability in water and the heptadiene–dione moiety that determines the physiochemical properties and antioxidant activity of curcumin is cleaved at high temperatures [33]. Therefore, bioactivity changes in curcumin may only occur when a high temperature causes the cleavage of the heptadiene–dione moiety. Hence, a higher temperature (150–200 °C) and prolonged duration (30 min) of oven heating was used for assessing the MEC heat resistance.

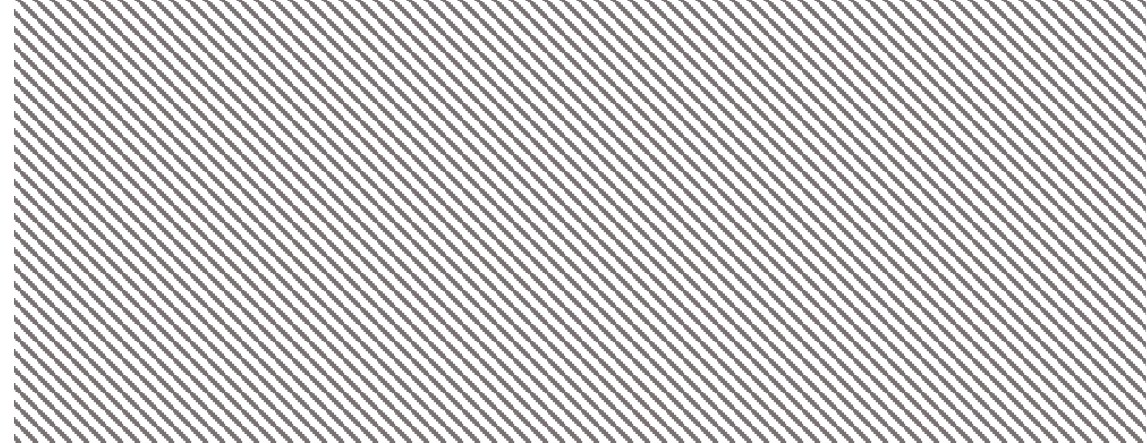

**Figure 2.** 2,2-diphenyl-1-picrylhydrazyl free radical (DPPH) free radical scavenging activities of MECs after water bath at different temperatures. MECs were reconstituted by methanol to achieve the 50-ppm curcumin concentration. All values are means of triplicates. Means followed by capital letters are significantly different due to thermal treatments ($p < 0.05$). Means with different lowercase letters within the same set temperature group are significantly different caused by samples ($p < 0.05$).

### 3.3.3. Changes in DPPH Free Radical Scavenging Activity Due to Oven Heating

Because no significant difference was noted in DPPH free-radical scavenging activity at 50–100 °C water bath heating between the S7–S9 and the control ($p \geq 0.05$), this experiment aimed to use higher temperatures (150–200 °C) and prolonged heating duration (30 min) for investigating the effects on DPPH free radical scavenging activity in the different samples.

Figure 3 presents the changes in DPPH free radical scavenging activity under different oven heating temperatures at a concentration of 50 ppm in the samples. For the control sample, data were only collected up to 160 °C; as the applied oven temperature was higher than 160 °C, the sample became seriously collapsed with obvious case hardening and burnt smell. This finding may be explained by the visual observation of natural curcumin being completely damaged after 30 min of oven heating at 170 °C, melting into a dark brown and viscous turmeric oil-like substance. This is consistent with previous research showing that the intramolecular bonds of curcumin break under high temperatures, resulting in the exposure of polar site and causing particle melting [33].

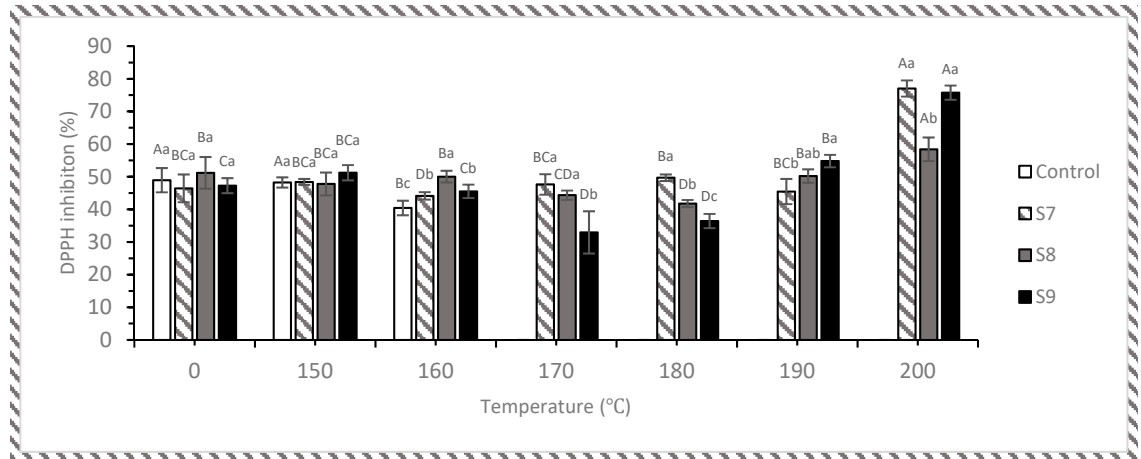

**Figure 3.** DPPH free radical scavenging activities of MECs after oven heating at different temperatures. MECs were reconstituted by methanol to achieve the 50-ppm curcumin concentration. All values are means of triplicates. Means followed by capital letters are significantly different due to thermal treatments ($p < 0.05$). Means with different lowercase letters within the same set temperature group are significantly different caused by samples ($p < 0.05$).

Therefore, in this experiment, we only investigated the natural sample and MEC samples at temperatures between 0–160 °C; the results showed that the scavenging activity in the control at 0 °C and 150 °C was 48.94% ± 3.72% and 48.24% ± 1.56%, respectively, and that there was no significant difference ($p \geq 0.05$). This finding demonstrates that natural curcumin is relatively stable when heated for 30 min at 0–150 °C. When temperature was increased to 160 °C, there was a significant decrease in the scavenging activity, which was only 40.42% ± 2.23%—a 17.23% decrease compared with the unheated samples ($p < 0.05$). This finding is consistent with the abovementioned finding that the heptadiene–dione group cleaves under high temperatures, resulting in a decrease in antioxidant activity [33].

Although S7 showed a decrease in scavenging activity at 160 °C (44.13% ± 1.14%), it increased after further increase in temperature, with the other MEC samples also showing similar trends: S8 showed a decrease at 170 °C and 180 °C, with scavenging activities of 44.34% ± 1.43% and 41.78% ± 1.09%, respectively, and S9 showed decreasing trends at the same temperatures, with scavenging activities of 32.94% ± 6.48% and 36.42% ± 2.18%, respectively. However, S8 and S9 were similar to S7 and their scavenging activities increased after further increase in temperature. At the highest temperature (200 °C), the scavenging activities of the three samples were the highest (77.04% ± 2.48%, 58.40% ± 3.62%, and 75.76% ± 2.19%, respectively, for S7, S8, and S9).

The trend of decrease in scavenging activity and then increase was also observed during water bath heating, and the decreasing trend was not positively correlated with temperature. Therefore, this decreasing phenomenon can be explained by the tight binding between the mesh structure of wall material and core material (curcumin) or by the chemical changes between the two, which increased the scavenging activity. According to the literature, when temperature is increased, the intermolecular hydrogen bonds of curcumin break, which increases curcumin solubility. In addition, high temperatures break other chemical bonds, causing the exposure of polar sites (-OH) and ketone groups (C=O), further increasing the solubility [33]. Therefore, additional increases in temperature disrupt the mesh structure of the wall material and simultaneously break the chemical bonds in curcumin and increase its solubility. This event causes its DPPH free radical scavenging activity to increase under the same concentrations. Furthermore, our experiment indirectly proves that microencapsulation exerts some protective effects. We observed that the free radical scavenging activity of the control began to substantially decrease at 160 °C and that the particles were completely damaged at 170 °C through visual observation. However, the DPPH free radical scavenging activity of MEC samples (S7–S9) increased at the highest temperature (200 °C) and reached its maximum value, demonstrating that the core temperature of MEC-encapsulated curcumin did not exceed 170 °C when an oven heating temperature of 200 °C was used.

### 3.3.4. Changes in ABTS Free Radical Scavenging Activity Due to Water Bath

Figure 4 shows changes in the ABTS free radical scavenging activity when samples undergo heating under different water bath temperatures; the control is a non-MEC sample. Similar to the DPPH free radical scavenging activity results, there was no significant difference in ABTS free radical scavenging activity at 0–100 °C ($p \geq 0.05$), showing that the ABTS free radical scavenging activity of curcumin is relatively stable at 0 °C and 50–100 °C.

Figure 4 also indicates the ABTS free radical scavenging activity of S7–S9. The ABTS free radical scavenging activity of S8 decreased at 90 °C (11.24% ± 2.36%), whereas there were no significant differences at the remaining temperatures ($p \geq 0.05$). S7 showed a higher ABTS scavenging activity among the MEC samples, which increased with the increase in temperature: the ABTS free radical scavenging activity increased by approximately 88.63% 0 °C and from 50 °C to 100 °C. Furthermore, S9 shows a similar trend as the ABTS free radical scavenging activity increased by 26.88% from 0 °C to 100 °C. This phenomenon can be explained by the results of previous research. As curcumin is relatively stable in an aqueous solution at 0–100 °C, the increasing temperatures break the intramolecular hydrogen bonds and expose the polar and ketone groups, which increases curcumin solubility, thereby

increasing the ABTS free radical scavenging activity [22,33]. The experimental HMG time of S7 is the central point (0) and also the median between those of S8 and S9. The incorporated excipient level of S7 is set at the lower end (−1) whereas HMG temperature level of S7 is set at the higher end (1). A higher HMG temperature could result in a better ABTS free radical scavenging activity through the observation of MEC S7. It also echoes that upon employing the least amount of excipient would help promote the resulted ABTS free radical scavenging activity. The excipient serves as a substance formulated alongside the curcumin; it is often referred to as "filler", or "diluents" [22], therefore, the least possible amount of utilization would be an optimal option as long as the minimum possible ER being retained in a considerably high level [22].

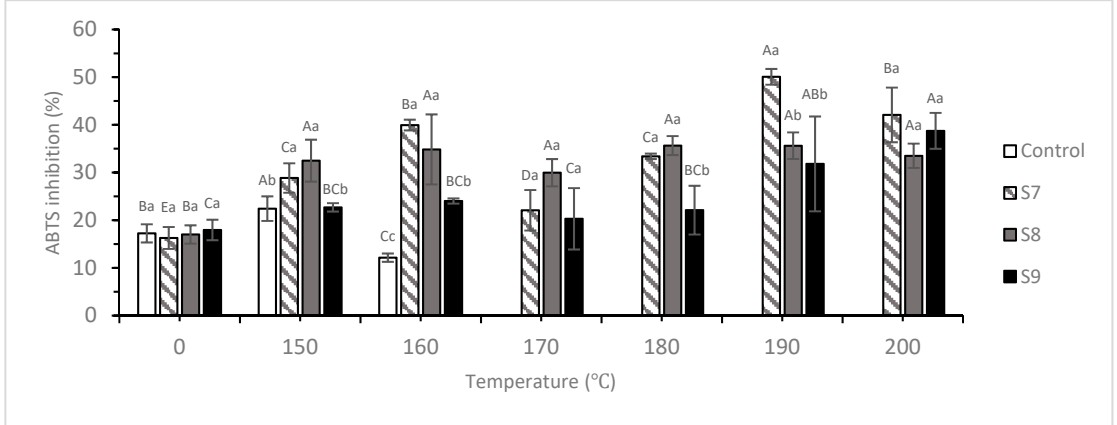

**Figure 4.** 2,2′-azino-bis(3-ethylbenzothiazoline-6-sulphonic acid (ABTS) free radical scavenging activities of MECs after water bath at different temperatures. MECs were reconstituted by methanol to achieve the 50-ppm curcumin concentration. All values are means of triplicates. Means followed by capital letters are significantly different due to thermal treatments ($p < 0.05$). Means with different lowercase letters within the same set temperature group are significantly different caused by samples ($p < 0.05$).

On comparing the results in Table 2, we observed that the ABTS free radical scavenging activity is poorer than DPPH free radical scavenging activity at the same concentration and that the DPPH free radical scavenging activity does not increase with temperature, whereas the ABTS free radical scavenging activity does. First, the DPPH free radical scavenging activity is better than the ABTS free radical scavenging activity at the same temperature because DPPH free radicals are non-polar (hydrophobic) substances, whereas ABTS is a polar (hydrophilic) substance. Under conditions of neutral pH, curcumin, being a non-polar (hydrophobic) substance, does not easily dissolve in water. Therefore, at the same concentrations, DPPH free radicals tend to be reduced by the hydrogen atoms in curcumin to form a stable molecular state, resulting in better scavenging activity. In addition, previous research has mentioned that high temperatures will lead to the exposure of polar groups and that ABTS free radicals are hydrophilic. Therefore, the ABTS free radical scavenging activity of curcumin is more sensitive to temperature changes compared with the DPPH free radical scavenging activity [33].

3.3.5. Changes in ABTS Free Radical Scavenging Activity Due to Oven Heating

Figure 5 shows changes in ABTS free radical scavenging activity in samples under different oven temperatures. In the control sample, the ABTS free radical scavenging activity significantly increased from 17.24% ± 1.90% to 22.43% ± 2.58%, when the temperature increased from 0 to 150 °C, which conforms to the phenomenon that the ABTS free radical scavenging activity increases with increase in temperature [33]. However, further increase in temperature to 160 °C resulted in a rapid decrease in the free radical scavenging activity to 12.15% ± 0.87%, demonstrating that the high temperature of 160 °C cleaves the heptadiene–dione moiety and decreases the antioxidant activity,

which conforms to expected results. In the MEC samples (S7–S9), it was observed that their free radical scavenging activity increased with temperature increases. However, the ABTS free radical scavenging activity of S7 decreased at 170 °C. Although this activity decreased in the remaining two MEC samples, there was no significant difference ($p \geq 0.05$). This result was similar to the abovementioned results and can be explained by the interaction between the microencapsulation wall material and curcumin.

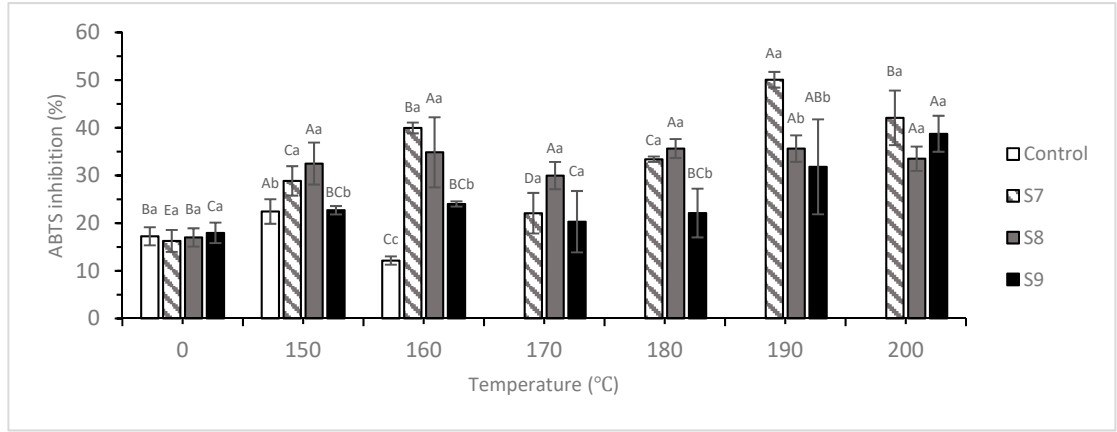

**Figure 5.** ABTS free radical scavenging activities of MECs after oven heating at different temperatures. MECs were reconstituted by methanol to achieve the 50-ppm curcumin concentration. All values are means of triplicates. Means followed by capital letters are significantly different due to thermal treatments ($p < 0.05$). Means with different lowercase letters within the same set temperature group are significantly different caused by samples ($p < 0.05$).

Comparison with the DPPH scavenging activity showed that the increase in the ABTS free radical scavenging activity in the MEC sample was greater at higher temperatures, with a mean increase of 125% from 0 to 200 °C. This proves the abovementioned hypothesis that high temperatures lead to the exposure of polar groups outside and increase the ABTS free radical scavenging activity [33,34].

From a series of heat resistance analyses, we observed that the curcumin content and antioxidant activity were not positively correlated, indicating that curcumin degradation does not necessarily lead to a decrease in the antioxidant activity. On the other hand, a suitable level of heating can aid in increasing the antioxidant activity of curcumin. Reportedly, the HPLC results of anthocyanin, a similar polyphenol, when heated at a range of 100–165 °C, showed that although anthocyanin was degraded, its antioxidant activity (determined by DPPH and ABTS assay) was not significantly affected [35]. However, this was considered to be attributable to the anthocyanin degradation products that maintain the antioxidant activity [36]. This phenomenon can explain the increased antioxidant activity of curcumin at high temperatures, despite it being degraded at these temperatures. In addition, Yang, Hsu and Yang [37] reported that the ferric-reducing antioxidant activity was increased when black soybean was roasted at 180 °C for 20 min. Zhou, Cai, and Xu [38] showed that the optimal roasting conditions was 210–230 °C for 30–35 min for yellow and black soybeans. Furthermore, Xu and Chang [39] reported that pressure steaming or steaming alone caused significant increases in the DPPH radical scavenging activity and in other antioxidant-related indices in yellow and black soybeans.

### 3.3.6. Color Changes

In the color system, brightness (L* value), redness (a* value), and yellowness (b* value) attributes are used. A higher brightness (0–100) indicates that the color is brighter, whereas a lower value indicates that the color is darker brown. A positive redness value indicates red color, whereas a negative value indicates green color. A positive yellowness value indicates yellow color, whereas a negative value indicates blue color, with a larger value indicating a darker color.

During oven heating, browning will occur. Therefore, it is expected that the L*value will decrease as color becomes darker. The browning reaction can be categorized into enzymatic and non-enzymatic. Enzymatic browning is mainly caused by polyphenol oxidase (PPO) activity, wherein it hydrogenates monohydric phenol into dihydric phenol before oxidation and further into diphenylquinone, followed by oxidative polymerization to form a black substance [40]. However, some studies have reported that PPO activity will decrease when heated at 80 °C for 30 min [41,42]. Heating to disrupt PPO activity is commonly used in the food processing industry to prevent enzymatic browning of polyphenols. Therefore, we can exclude that the present browning reaction is enzymatic in nature. Non-enzymatic browning can be divided into Maillard reaction (also known as amino-carbonyl reaction), caramelization, and ascorbic acid oxidation. Ascorbic acid oxidation mainly occurs in vegetable and fruit juices and can be prevented by fixation. Caramelization refers to the production of brown products under high temperature heating or acid–base treatment when carbohydrates are present in the absence of amino compounds [42–44]. Because curcumin MEC itself does not contain amino compounds, amino-carbonyl reaction can be excluded. Ascorbic acid oxidation can be disrupted by fixation. Therefore, the browning reaction was speculated to be due to caramelization. Caramelization typically occurs at temperatures higher than the melting point of the carbohydrate, and any carbohydrate can undergo caramelization [42].

The oven heating temperature and duration of the different samples are fixed and every sample was placed in the center of the oven to avoid any effect of temperature differences. Therefore, color variations because of manipulations can be avoided. A previous study reported that phosphatidylethanolamine (PE) and glucose undergo browning in a non-polar medium and that the product exhibits high antioxidant activity and maintains lipid oxidation stability as well as increases the stability of different free radical scavengers [34]. Therefore, this finding could be associated with the results of the DPPH and ABTS free radical scavenging activity as oven temperature increases.

Table 4 shows that all color indices significantly decreased at each temperature point in the control. In particular, a drastic decrease (L* value of 17.95 ± 0.06) was observed at 170 °C, along with the presence of dark brown color and complete destruction through visual observation. At 0 °C and 150–200 °C, L* value decreased by 68.99%, showing that the non-microencapsulated natural curcumin is extremely sensitive to heating at temperatures >150 °C and that particle begins to collapse when temperature is increased to 160 °C.

In addition, although we observed significant differences in the MEC samples, they were relatively stable compared with the control, with the decrease in L* value at 0–200 °C being 13.49–24.78%, demonstrating that curcumin MEC maintains some brightness under high-temperature oven heating.

From Table 4, we observed that the a* value in the control at the oven heating temperature of 150 °C did not greatly decrease; however, a further increase in temperature caused a substantial decrease in its a* value from 24.01% ± 0.43% to 1.07% ± 0.01%, showing that color changed from red to dark brown, thereby indirectly proving that caramelization occurred.

A similar situation was observed in the MEC samples, although the a* values increased when the temperature was increased to 190 °C. This phenomenon can be explained by the disintegration of MEC wall material owing to the high temperature, which results in the exposure of curcumin and, subsequently, in the presentation of the original color. Moreover, this finding reiterates the aforementioned result that high temperatures disrupt the mesh structure of the wall material and result in the complete release of curcumin, leading to an increase in the antioxidant activity.

The b* value of the non-microencapsulated samples substantially decreased under oven heating temperature of 150 °C from 34.65% ± 0.59% to 25.31% ± 0.04% and the color changed from yellow to dark brown. This finding was similar to the aforementioned results, proving the occurrence of caramelization. In the MEC sample, the b* value increased after high temperature oven heating (150–170 °C) and appears as golden-yellow. Further increase in temperature (180–190 °C) resulted in the Maillard reaction and caused the b* value to decrease, and bright yellow was gradually lost.

**Table 4.** Changes in color indices (L*, a*, and, b*) for MECs (5% of wall material contained in S7–S9) due to different oven heating temperatures.

| Index | Temperature (°C) | Sample | | | |
|---|---|---|---|---|---|
| | | Control | S7 | S8 | S9 |
| L* | 0 | 53.74 ± 1.03 $^{Ac}$ | 74.37 ± 0.53 $^{Cb}$ | 76.39 ± 1.34 $^{Ba}$ | 76.53 ± 0.38 $^{Ca}$ |
| | 150 | 40.81 ± 0.19 $^{Bc}$ | 77.57 ± 0.29 $^{Ab}$ | 77.41 ± 0.10 $^{Ab}$ | 78.69 ± 0.03 $^{Aa}$ |
| | 160 | 30.07 ± 0.55 $^{Cd}$ | 75.45 ± 0.24 $^{Bc}$ | 76.17 ± 0.18 $^{Bb}$ | 77.21 ± 0.02 $^{Ba}$ |
| | 170 | 17.95 ± 0.06 $^{Ed}$ | 73.06 ± 0.29 $^{Dc}$ | 75.19 ± 0.03 $^{Cb}$ | 75.60 ± 0.06 $^{Da}$ |
| | 180 | 18.63 ± 0.03 $^{Ed}$ | 71.14 ± 0.07 $^{Ec}$ | 73.63 ± 0.15 $^{Da}$ | 71.73 ± 0.22 $^{Eb}$ |
| | 190 | 19.75 ± 0.06 $^{Dd}$ | 67.10 ± 0.71 $^{Fb}$ | 72.21 ± 0.11 $^{Ea}$ | 64.75 ± 0.12 $^{Fc}$ |
| | 200 | 16.66 ± 0.06 $^{Fd}$ | 63.31 ± 0.25 $^{Gb}$ | 66.07 ± 0.24 $^{Fa}$ | 57.57 ± 0.17 $^{Gc}$ |
| a* | 0 | 24.82 ± 0.28 $^{Aa}$ | −1.40 ± 0.26 $^{Bc}$ | −0.82 ± 0.2 $^{Bb}$ | −2.40 ± 0.16 $^{Cd}$ |
| | 150 | 24.01 ± 0.43 $^{Ba}$ | −5.38 ± 0.09 $^{Dc}$ | −4.63 ± 0.11 $^{Cb}$ | −5.02 ± 0.25 $^{Dbc}$ |
| | 160 | 15.04 ± 0.53 $^{Ca}$ | −5.67 ± 0.05 $^{Eb}$ | −5.40 ± 0.28 $^{Db}$ | −6.60 ± 0.06 $^{Ec}$ |
| | 170 | 1.07 ± 0.01 $^{Fa}$ | −6.92 ± 0.09 $^{Fd}$ | −6.10 ± 0.18 $^{Eb}$ | −6.52 ± 0.13 $^{Ec}$ |
| | 180 | 1.78 ± 0.01 $^{Ea}$ | −5.83 ± 0.13 $^{Ec}$ | −6.97 ± 0.16 $^{Fd}$ | −5.10 ± 0.11 $^{Db}$ |
| | 190 | 5.35 ± 0.01 $^{Da}$ | −3.93 ± 0.22 $^{Cc}$ | −4.63 ± 0.06 $^{Cd}$ | −0.87 ± 0.14 $^{Bb}$ |
| | 200 | 2.23 ± 0.39 $^{Eb}$ | 0.13 ± 0.06 $^{Ac}$ | −0.34 ± 0.13 $^{Ad}$ | 3.02 ± 0.08 $^{Aa}$ |
| b* | 0 | 34.65± 0.59 $^{Aa}$ | 25.83 ± 0.09 $^{Fb}$ | 22.81 ± 0.55 $^{Gd}$ | 24.05 ± 0.29 $^{Fc}$ |
| | 150 | 25.31 ± 0.04 $^{Bd}$ | 35.67 ± 0.04 $^{Ca}$ | 34.01 ± 0.04 $^{Db}$ | 33.27 ± 0.02 $^{Cc}$ |
| | 160 | 12.83 ± 0.29 $^{Cd}$ | 38.73 ± 0.29 $^{Aa}$ | 37.79 ± 0.02 $^{Ab}$ | 36.94 ± 0.04 $^{Ac}$ |
| | 170 | 0.11 ± 0.12 $^{Fd}$ | 37.62 ± 0.12 $^{Fd}$ | 37.38 ± 0.01 $^{Bb}$ | 35.29 ± 0.05 $^{Bc}$ |
| | 180 | 0.67 ± 0.04 $^{Ed}$ | 34.24 ± 0.04 $^{Ed}$ | 34.89 ± 0.08 $^{Ca}$ | 31.74 ± 0.07 $^{Dc}$ |
| | 190 | 2.32 ± 0.03 $^{Dc}$ | 30.03 ± 0.03 $^{Dc}$ | 29.28 ± 0.02 $^{Ea}$ | 26.24 ± 0.04 $^{Eb}$ |
| | 200 | 2.74 ± 0.46 $^{Dd}$ | 25.24 ± 0.46 $^{Dd}$ | 24.27 ± 0.04 $^{Fb}$ | 22.48 ± 0.03 $^{Gc}$ |

All values are means of triplicates. Means bear capital letters within the same column are significantly different due to temperature ($p < 0.05$). Means bear lowercase letters in the same row are significantly different due to samples ($p < 0.05$).

Color analysis results showed that under the heating temperatures of >150 °C, Maillard reaction occurs in the non-microencapsulated sample, causing its color to rapidly change to dark brown and the visual observation of the particles to be gradually disrupted. When the non-microencapsulated sample was heated to 170 °C, particles were completely destroyed to become an oil-like substance. In comparison, some degree of color change occurred in the microencapsulated sample. This was owing to the absence of red-brown or dark brown color caused by the Maillard reaction from direct flames. This finding reflects the protective function of microencapsulation under oven heating. In addition, if the heating temperature was controlled at 150–170 °C, MECs would appear in the preferred golden-yellow color. The caramelization reaction mentioned here may reflect the increase in antioxidant activity mentioned previously [33].

### 3.4. Correlation Analysis and Practical Justification

Table 5 presents Pearson's product moment correlation coefficient between the antioxidant abilities and color parameters. Generally, negative correlations between L*/b* and both free radical scavenging abilities were obtained regardless sample variation (S7–S9). However, a nearly positive correlation between a* and both free radical scavenging abilities were obtained regardless sample variation (S7 and S9) A significant correlation between color indices and ABTS free radical scavenging ability for S9; a strong positive correlation was observed for a* whereas strong negative correlations was observed for L* and b*. A positive correlation between a* and DPPH free radical scavenging ability was also observed especially with S7 and S8. We also noted that there were significant differences ($p < 0.05$), curcumin content was relatively stable between 0 °C and 50–100 °C of the hydrothermal process; this finding is consistent with the aforementioned antioxidant results. Additionally, the curcumin content in non-MEC control and S7–S9 samples significantly decreased as the temperature increased

while the oven heating was applied. In contrast to the aforementioned antioxidant activity, curcumin content did increase under high temperature, indicating that the recovery of antioxidant activity at high temperature is correlated to the tight binding between curcumin and wall material. Therefore, the increase in MEC antioxidant activity during oven heating was owing to disruption of the chemical bonds by the high temperature, which exposes the certain functional groups, thereby increasing the solubility and caramelization; moreover, the glucose reaction products produced during oven heating can increase the antioxidant activity [32,33].

**Table 5.** Pearson correlation analysis between antioxidant abilities and color parameters for oven heating ranging from 150–200 °C.

| Activities | Sample | Correlation Coefficient ($R^2$) | | |
| --- | --- | --- | --- | --- |
| | | L* | a* | b* |
| **DPPH** | S7 | −0.714 | 0.889 [†] | −0.800 |
| | S8 | −0.705 | 0.952 [‡] | −0.777 |
| | S9 | −0.764 | 0.628 | −0.817 [†] |
| **ABTS** | S7 | −0.601 | 0.597 | −0.600 |
| | S8 | −0.197 | −0.031 | −0.241 |
| | S9 | −0.930 [‡] | 0.972 [‡] | −0.918 [‡] |

[†] $p < 0.05$, [‡] $p < 0.01$.

We also noticed that maybe it was more useful at room temperature as control condition instead of 0 °C for free radical scavenging activity and color retention evaluations. However, the chosen control temperature (0 °C) followed some other previous studies [22,34]. Additionally, the exposure of microcapsules to room temperature for thermal equalization would cause a certain degree of moisture rehydration. We would also report that the reconstituting solvent (methanol), the thermal stability and the antioxidant property of MEC after spray drying are not sufficient to use these materials in foods even though some previous studies can be considered as supportive references. It has been reported that the ternary wall material matrices of maltodextrin, gum Arabic, and modified starch as wall material for curcumin encapsulation by spray drying was effective to prevent loss of curcumin and colour changes after an 8-week storage under incident light [17]. A study also indicated that curcumin-based hydrophilic colorants obtained by assisted by spray-drying possessed a wide range of pH and heat stability while k-carrageenan, poly (vinyl alcohol), and polyvinylpyrrolidone was used as encapsulant materials [24].

### 4. Conclusions

The highest minimum possible encapsulation rate when porous starch was used as spray drying MEC wall material was 52.23%. The DPPH free radical scavenging activity of encapsulated curcumin was relatively stable under water bath heating and reached its maximum value under oven heating (77.04 ± 2.48). The DPPH free radical scavenging activity of encapsulated curcumin was better than its ABTS free radical scavenging activity. The non-microencapsulated sample underwent rapid browning reaction under oven heating temperatures of >160 °C and its color appeared dark brown. Under high oven heating temperatures, the color of MEC was well maintained and appeared golden-yellow when temperature was maintained at 150–170 °C. The study results showed that MEC can effectively protect bioactivity and color under high oven heating temperatures and that the antioxidant activity of curcumin increases after oven heating. The use of β-cyclodextrin and porous starch as MEC wall material and use of spray drying to prepare MEC would feasible for the bakery industry with the pH stability, the solubility, the interaction with the food compounds, and proper solvent system carefully evaluated.

**Author Contributions:** Conceptualization, H.Y.; Methodology, H.Y., C.H. and S.W.; Validation, H.Y., C.H. and S.W.; Formal Analysis, C.H. and S.W. Investigation, H.Y., C.H. and S.W.; Resources, H.Y.; Data Curation, H.Y.,

C.H. and S.W.; Writing—Original Draft Preparation, H.Y. and S.W.; Writing—Review & Editing, H.Y. and C.H.; Visualization, H.Y. and S.W.; Supervision, H.Y.; Project Administration, H.Y.; Funding Acquisition, H.Y. All authors have read and agreed to the published version of the manuscript.

**Funding:** This research was funded by the Department of Food Science, National Chiayi University, Taiwan, Republic of China.

**Acknowledgments:** The authors acknowledge the administrative and technical support provided by the Department of Food Science, National Chiayi University, Taiwan, Republic of China.

**Conflicts of Interest:** The authors declare no conflict of interest.

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
