# Peer review of "Evaluation of Thermal Effects on the Bioactivity of Curcumin Microencapsulated with Porous Starch-Based Wall Material Using Spray Drying"

_processes, doi:10.3390/pr8020172_

Round 1

Reviewer 1 Report

Comments for the authors:

Introduction must be improved by including more recent references and research findings on the proposed approach and by presenting the novelty of the study.

Line 72 - Mention the source from where the authors got the sample.

In Materials and Methods section, the authors should mention the properties of the encapsulation in detail in a separate section.

The experimental design must be improved.

Mention the bioactivity of the encapsulated Curcumin with a graphical representation.

Can Differential Scanning Calorimetry (DSC) and x-ray Diffraction methods be performed for the current study? Did the authors carry out these techniques?

Show the results of the thermal analysis—DSC and TGA by a graph.

Add Curcumin and microcapsule preparations stability.

Describe about the limitations and disadvantages of the proposed study.

The figures must be enlarged and their quality must be improved.

Reviewer 2 Report

The comments are reported in the word file.

Reviewer 3 Report

The manuscript entitled "Evaluation of thermal effects on the bioactivity of
curcumin microencapsulated with porous starch- based wall material using spray drying" deals with a topic of interest in the scientific community and it's experimental plan is well planned. However the English language is not always clear and the results are not always well discussed. 

The detailed comments are reported in the attached files. 

I suggest major revisions. 
